# The diagnostic value of bronchoalveolar lavage fluid metagenomic next-generation sequencing in critically ill patients with respiratory tract infections

Xiaohang Hu,[1] Liqing Jiang,[1] Xiaowei Liu,[2] Hong Chang,[1] Haixin Dong,[1] Jinyan Yan,[1] Xiaoya Zhou,[3] Min Kong[3]

ABSTRACT   Metagenomic next-generation sequencing (mNGS) is an unbiased and rapid method for detecting pathogens. This study enrolled 145 suspected severe pneumonia patients who were admitted to the Affiliated Hospital of Jining Medical University. This study primarily aimed to determine the diagnostic performance of mNGS and conventional microbiological tests (CMTs) using bronchoalveolar lavage fluid samples for detecting pathogens. Our findings indicated that mNGS performed significantly higher sensitivity (97.54% vs 28.68%, $P < 0.001$), coincidence (90.34% vs 35.17%, $P < 0.001$), and negative predictive value (80.00% vs 13.21%, $P < 0.001$) but performed lower specificity than CMTs (52.17% vs 87.5%, $P < 0.001$). *Streptococcus pneumoniae* as the most common bacterial pathogen had the largest proportion (22.90%, 30/131) in this study. In addition to bacteria, fungi, and virus, mNGS can detect a variety of atypical pathogens such as *Mycobacterium tuberculosis* and *non-tuberculous*. Mixed infections were common in patients with severe pneumonia, and bacterial-fungal-viral-atypical pathogens were the most complicated infection. After adjustments of antibiotics based on mNGS and CMTs, the clinical manifestation improved in 139 (95.86%, 139/145) patients. Our data demonstrated that mNGS had significant advantage in diagnosing respiratory tract infections, especially atypical pathogens and fungal infections. Pathogens were detected timely and comprehensively, contributing to the adjustments of antibiotic treatments timely and accurately, improving patient prognosis and decreasing mortality potentially.

IMPORTANCE   Metagenomic next-generation sequencing using bronchoalveolar lavage fluid can provide more comprehensive and accurate pathogens for respiratory tract infections, especially when considering the previous usage of empirical antibiotics before admission or complicated clinical presentation. This technology is expected to play an important role in the precise application of antimicrobial drugs in the future.

KEYWORDS   metagenomic next-generation sequencing, bronchoalveolar lavage fluid, severe pneumonia, respiratory tract infections, conventional microbiological tests

O f all infectious disease categories, respiratory tract infections (RTIs) impart the greatest mortality both worldwide and in the United States, which are extremely frequent in both adults and children, representing an increased economic burden, morbidity, and mortality (1). There are also potential pathogens in the lungs of asymptomatic population, which can become pathogenic pathogens for patients with low immunity (2). In China, the study showed that about 50% of the patients with RTIs, especially mixed infections, failed to be identified, which reduced the cure rate and increased the mortality rate (3). Various pathogens, including bacterial, fungal, viral, parasitic, and some atypical pathogens, had been reported to be the etiology of RTIs (4–9). With the report and recognition of coronavirus disease (COVID-19) in December 2019

Editor Benjamin M. Liu, Children's National Hospital, George Washington University, Washington, DC, USA

Ad Hoc Peer Reviewer Kaan Çeylan, University of Gaziantep, Gaziantep, Turkey

Address correspondence to Min Kong, zaozhuang.love@163.com.

The authors declare no conflict of interest.

in the city of Wuhan in Hubei province, the role of respiratory virus had been becoming more dominant (10). Liu et al. (11) found that primary immunodeficiency patients may be susceptible to SARS-CoV-2 infection after literature review. Infections which are caused by atypical pathogens such as *Pneumocystis jirovecii* and *Chlamydia psittaci* are often ignored by clinicians due to their unclear clinical symptoms, and such pathogens are difficult to be detected by conventional microbiological tests (CMTs) (12).

Bronchoalveolar lavage fluid (BALF), as a favorable specimen for RTIs, has been widely applied for the RTI diagnosis. Currently, traditional etiological detection methods are commonly applied for the diagnosis of RTIs such as CMTs and immunological tests, but they have poor timeliness, low pathogen coverage, and low positive detection rate, which makes it difficult to meet the diagnostic needs of critically ill patients (13). EPICIII studies (14) showed that 43%–60% of patients in the ICU was diagnosed with suspected or confirmed infections, but only 65% of these patients was positive for microbial culture, and it was difficult to determine whether the results had clinical significance due to contamination by colonizing bacteria easily. Another sample, such as blood, is also chosen as an alternative sample, but the accuracy is lower; a study showed that only 0%–14% of patients with severe respiratory infections had a positive detection rate in blood culture (15). The window period of serum antibody detection is difficult to be predicted accurately (16). PCR amplification can involve numerous individual tests for specifically targeted organisms due to the specific primers or probes, but it may fail to detect rare pathogens, and the application of mismatched primers to pathogens will decrease the sensitivity of detection (17), which is a challenge to the expertise of clinicians. The complexity of etiology means that empirical antibiotics will be used inevitably. Thus, a new, timely, and accurate multiplexed diagnostic method for detecting pathogens is necessary.

Metagenomic next-generation sequencing (mNGS) is a high-throughput sequencing technology, which has the advantages of unculture, unbias, unhypothesis, wide coverage, high sensitivity, and so on (18). It can detect almost all pathogens from the clinical samples, especially for rare, novel, unknown, and atypical pathogen infectious diseases (19, 20). Because of significant advantages for detecting atypical and slow-growing microorganisms, it is increasingly used in the clinical diagnosis of patients with mixed infections (18, 21). It can be applied for infections in all systems such as respiratory, central nervous, and blood systems (22–24). Compared with CMTs, it is less affected by the application of antibiotics in patients (25, 26) and has considerable advantages for rapid infection diagnosis, promoting targeted antimicrobial therapy, and improving patient prognosis. Huang et al. (27) detected 240 patients with suspected pulmonary infections using mNGS; the results showed that the positive rate of mNGS for patients with pulmonary infections (88.30%) was much higher than that of CMTs (25.73%). Therefore, identification and characterization of pathogenic microorganisms are crucial. However, few studies have investigated the performance of mNGS in diagnosing severe pneumonia of critically ill patients using BALF samples.

This is a retrospective study. A total of 145 patients with suspected severe pneumonia were enrolled in our hospital. BALF samples obtained from the locus of infection were detected for CMTs and mNGS. Furthermore, by comparing CMTs with mNGS and analyzing the survival curve, the diagnostic performance of mNGS using BALF in critically ill patients with RTIs was evaluated.

## RESULTS

### Basic characteristics

A total of 145 patients who were admitted to the respiratory critical care and intensive care unit were enrolled according to the strict enrollment criteria in this retrospective study. Out of all patients, 109 (75.17%) were male and the average age was 63.6 years. The average hospital stay was 20 days, ranging from 1 to 77 days. A total of 92 patients had underlying diseases, including 35 hypertension, 26 cerebrovascular disease, 25 cardiovascular disease, 18 diabetes, 13 lung disease, blood disease, chronic liver disease,

and tumor after chemotherapy. Some patients may have multiple underlying diseases. In clinical comprehensive diagnosis, 131 patients were finally diagnosed with severe pneumonia, including 55 monobacterial infections and 76 mixed infection patients. The average age of the mixed infections group was higher than that of the monobacterial infections group, and there was significant statistical differences between the two groups. The percentage of neutrophil, lymphocyte, and Glactomannan (GM) antigen had significant statistical difference, but the white blood cell (WBC) count and serum CRP had no significant statistical difference between the two groups. Finally, 110 patients (75.86%, 110/145) were cured, 29 patients (20.00%, 29/145) experienced improved conditions, and 6 patients (4.14%, 6/145) died. The cure rate and 3-month survival rate of the monobacterial infections group were higher than those of the mixed infections group (Fig. 1). The baseline of all enrolled patients is shown in Table 1.

## Comparison of mNGS and CMTs

In this study, a total of 404 pathogens were detected from all enrolled patients by mNGS, consisting of 299 bacteria, 54 viruses, 60 fungi, and other atypical pathogens. The pathogen spectrum revealed that bacteria were the most common pathogens, of which the leading bacteria were *S. pneumoniae* (10.00%, 30/300), *Streptococcus constellation* (7.33%, 22/300), *Parvimonas micra* (7.00%, 21/300), *Enterococcus faecium* (5.33%, 16/300), and *Klebsiella pneumoniae* (5.00%, 15/300). A total of 54 fungi were identified, of which the most frequent was *Candida albicans* (40.74%, 22/54). Of 60 viruses, the most common was *Human gamma herpesvirus type* 4 (EBV) (35.00%, 21/60), followed by *Human alpha herpesvirus type* 1 (HSV1) (33.33%, 20/60). A number of atypical pathogens included *Actinomyces israelii*, *Mycobacterium tuberculosis*, *Mycoplasma hominis*, and *C. psittaci*. Only 72 pathogens were identified by CMTs, which were mainly Gram-negative bacteria, of which *Acinetobacter baumannii* (18.06%, 13/72) was the most common bacteria and *C. albicans* (12.5%, 9/72) was the most common fungi. In addition, the positive rate of mNGS for all detected pathogens was significantly higher than that of CMTs ($P < 0.01$). The positive rate of CMTs for *Pseudomonas aeruginosa* was higher than that of mNGS, but other bacteria showed a lower positive rate in the monobacterial infections group. The positive rate of CMTs for all pathogens was lower than that of mNGS in the mixed infections group. The detected number of *Dialister pneumosintes* and that of*Streptococcus constellatus* were higher in the monobacterial infections than that in the

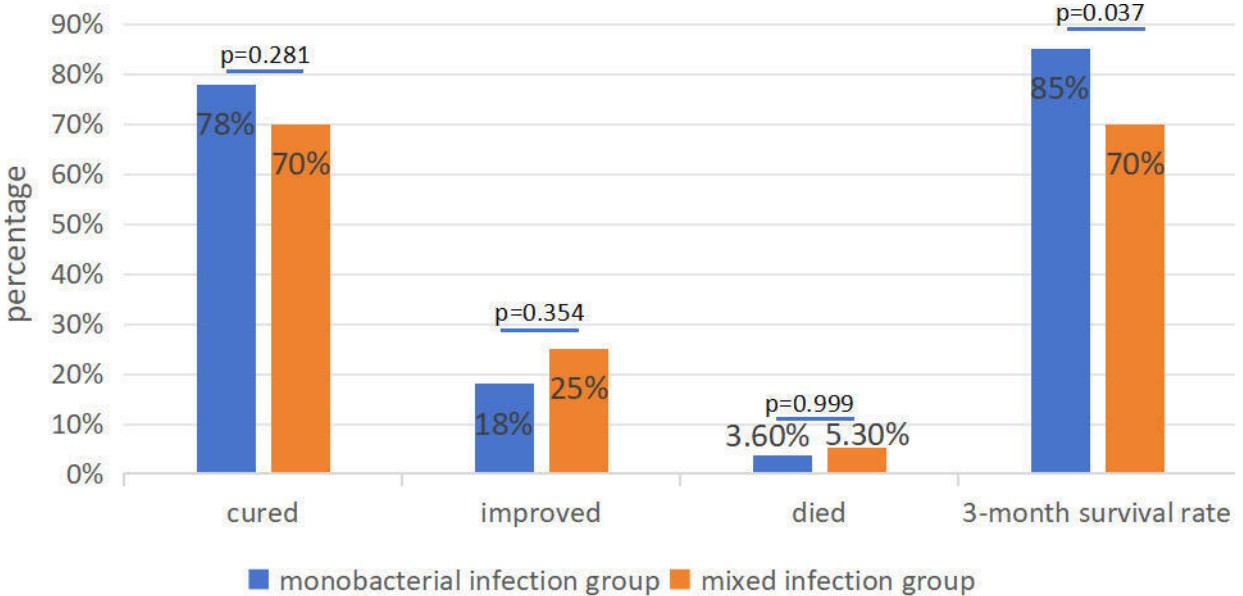

FIG 1 Treatment outcomes of patients with monobacterial infections and mixed infections, the cure rate, and the 3-month survival rate of the monobacterial infections group were higher than those of the mixed infections group; the 3-month survival rate had statistical difference between the two groups.

**TABLE 1** Baseline of 145 patients enrolled in this study

| Characteristic | Total ($n$ = 145) | Monobacterial infections group ($n$ = 55) | Mixed infections group ($n$ = 76) | $P$ value |
|---|---|---|---|---|
| Male, $n$ (%) | 109 (75.17%) | 46 (83.63%) | 57 (75.00%) | 0.234 |
| Age, years, means ($Q_1$, $Q_3$) | 63. 6 (57, 75) | 60. 5 (52, 71) | 67. 00 (61, 76) | 0.030 |
| Comorbidities [$n$ (%)] | | | | |
| Hypertension | 35 (24.14%) | 12 (21.82%) | 23 (30.26%) | 0.103 |
| Cerebrovascular disease | 26 (17.93%) | 11 (20.00%) | 15 (19.74%) | 0.555 |
| Cardiovascular disease | 25 (17.24%) | 6 (10.91%) | 19 (25.00%) | 0.010 |
| Diabetes | 18 (12.41%) | 5 (9.09%) | 13 (17.11%) | <0.001 |
| Pulmonary disease | 13 (8.97%) | 9 (16.36%) | 4 (5.26%) | 0.104 |
| Immune dysfunction | 7 (4.83%) | 3 (5.45%) | 4 (5.26%) | 0.802 |
| Malignant tumor after Chemoradiotherapy | 6 (4.14%) | 2 (3.64%) | 4 (5.26%) | 0.480 |
| Chronic liver diseases | 5 (3.45%) | 2 (3.64%) | 3 (3.95%) | 0.735 |
| Other diseases | 10 (6.90%) | 6 (10.91%) | 4 (5.26%) | 0.419 |
| Length of stay, days, means ($Q_1$, $Q_3$) | 20 (12, 27) | 18. 4 (13, 25) | 22 (14, 31) | 0.082 |
| Outcomes [$n$ (%)] | | | | |
| Cure | 110 (75.86%) | 43 (78.18%) | 53 (69.74%) | 0.284 |
| Improvement | 29 (20.00%) | 10 (18.18%) | 19 (25.00%) | 0.354 |
| Death | 6 (4.14%) | 2 (3.64%) | 4 (5.26%) | 0.999 |
| Three-month survival rate | 115 (79.31%) | 47 (85.45%) | 53 (69.74%) | 0.037 |
| Laboratory examination (mean) | | | | |
| White blood cell (×109/L) | 10.76 | 11.20 | 10.45 | 0.716 |
| Neutrophil (%) | 79.75 | 76.06 | 82.37 | <0.001 |
| Lymphocyte (%) | 12.68 | 14.59 | 11.33 | 0.009 |
| C-reactive protein (mg/L) | 100.76 | 93.35 | 105.92 | 0.693 |
| GM antigen | 0.50 | 0.22 | 0.68 | <0.001 |
| G test | 94.00 | 82.18 | 102.14 | 0.257 |

mixed infections, and the number of other bacteria was lower. The detailed information on pathogens that meet the infection definition by mNGS and CMTs is shown in Fig. 2.

In this study, the comparison between mNGS and CMTs indicated that mNGS showed higher diagnostic sensitivity and coincidence, which were significantly higher than those of CMTs (97.54% vs 28. 68%, $P < 0.001$; 90.34% vs 35.17%, $P < 0. 001$), but the specificity of mNGS was lower than that of CMTs (52.17% vs 87.5%, $P < 0.001$). The positive predictive value (PPV) of mNGS and CMTs showed no significant difference (91.54% vs 94.87%, $P = 0.390$), but the negative predictive value (NPV) of mNGS was significantly higher than that of CMTs (80.00% vs 13.21%, $P < 0.001$) (Fig. 3; Table 2). Out of the 145 suspected pneumonia patients, there were 39 cases (26.90%) with only culture positive and 130 cases (89.66%) with only mNGS positive, respectively. A total of 93 cases (64.14%, 93/145) were positive for pathogens detected by mNGS only, of which 12 cases were false positive. Two cases (1.38%, 2/145) were positive by CMTs only, of which one case was false positive. Thirteen cases (8.97%, 13/145) were found negative by both methods, of which two cases were false negative (Fig. 4A through C). Moreover, out of all enrolled patients, the concordant results of mNGS and CMTs were 50 samples (37 double positive and 13 double negative), with a matching rate of 34.48% (50/145). Among the 37 double-positive cases, only 5 cases were completely matched and 2 cases were totally mismatched (CMTs showed *P. aeruginosa* and *S. pneumoniae*, but mNGS showed fungi) and the remaining 30 cases were partially matched (Fig. 4D). mNGS detected some pathogens that were difficult to be detected by CMTs, including *Aspergillus*, *Rhizopus*, *Haemophilus*, *M. tuberculosis*, *Mycoplasma*, and various viruses.

All the patients received empirical antibiotic treatment before or after admission. The positive rate and coincidence rate of mNGS were significantly higher than those of CMTs (96.36% vs 20.00%; 89.09% vs 21.82%) in 55 monobacterial infections. Among the 76 mixed infections, bacteria-virus (39.47%, 30/76) was the most common, followed by

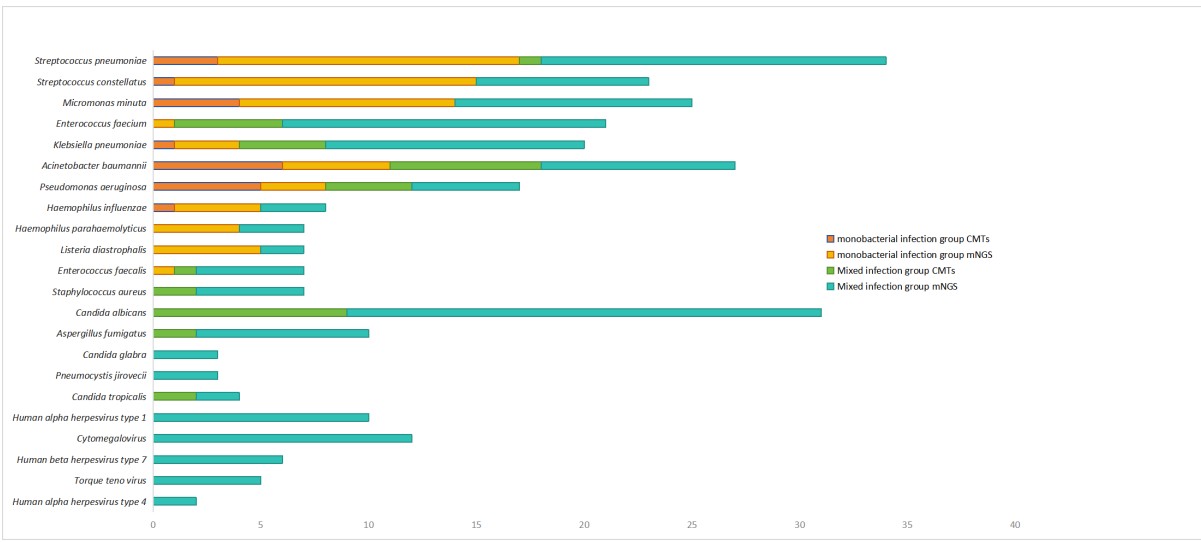

**FIG 2** Overlap of main detected pathogens of mixed and monomicrobial infections using mNGS and CMTs. Detection efficiency of mNGS and CMTs for specific pathogens is different.

bacteria-fungi-virus (32.89%, 25/76) and bacteria-fungi (27.64%, 21/76), The positive and coincidence rates of mNGS were significantly higher than those of CMTs (100.00% vs 32.89%; 93.42% vs 32.89%). The positive rate and coincidence rate of mNGS in mixed infections were higher than those in monobacterial infections. These data indicate that mixed infections are a common type of severe pneumonia and mNGS has obvious advantages in the diagnosis of mixed infections and seems less likely to be influenced by the empirical antibiotics treatments than CMTs before sampling.

## The influence of mNGS on diagnosis and antibiotic treatment

One hundred thirty-one cases were diagnosed with severe pneumonia, and 29 (20.00%, 29/145) cases were diagnosed by mNGS only, while only 1 case was diagnosed by CMTs. Besides the results of CMTs and mNGS, the remaining 101 cases were comprehensively diagnosed based on multiple clinical factors, including the clinical manifestations, epidemiology, laboratory results, imaging results, and treatment outcomes. Besides this, the turnaround time of mNGS was about 2 days, while that of CMTs was about 5 days. In terms of treatment, of 55 monobacterial infection cases, 14 cases were adjusted according to mNGS, 11 cases were cured, and 5 cases were replaced with tigecycline because of *A. baumannii* (3 cases were discharged with improvement, 1 case died due to cerebral infarction, and 1 case died after 3 months of active discharge). Four cases were adjusted to cephalosporins or quinolones because of *S. pneumoniae*, and all the conditions improved. One case was adjusted to moxifloxacin and minocycline because of *Streptococci*, *Nutrient deficient strain*, and *Chlamydia psittaci*, and the condition improved after treatment. One case was dosely increased with the anti-tuberculosis drugs including isoniazid, pyrazinamide, and rifampicin due to the detection of *M. tuberculosis* but died due to severe infection. Among the 76 mixed infections, 25 cases were adjusted according to mNGS, while the rest abandoned adjustment due to the coverage of previous antibiotics. Seventeen cases were cured, of which 11 cases were treated with antifungal drugs including voriconazole, fluconazole, or caspofungii due to the detection of *Candida* or *Aspergillus* (eight patients were discharged, two cases gave up treatment due to no obvious improvement of symptoms and died 3 months later, and one case died due to B-lymphoblastoma). Three cases showed rapid improvement after increasing ganciclovir due to detection of EBV, and two cases were adjusted to penicillin G with detection of *Actinomyces graevenitzii* or *S. pneumoniae*. One case was discharged after being modified to minocycline due to the detection of *C. psittaci*. One case was

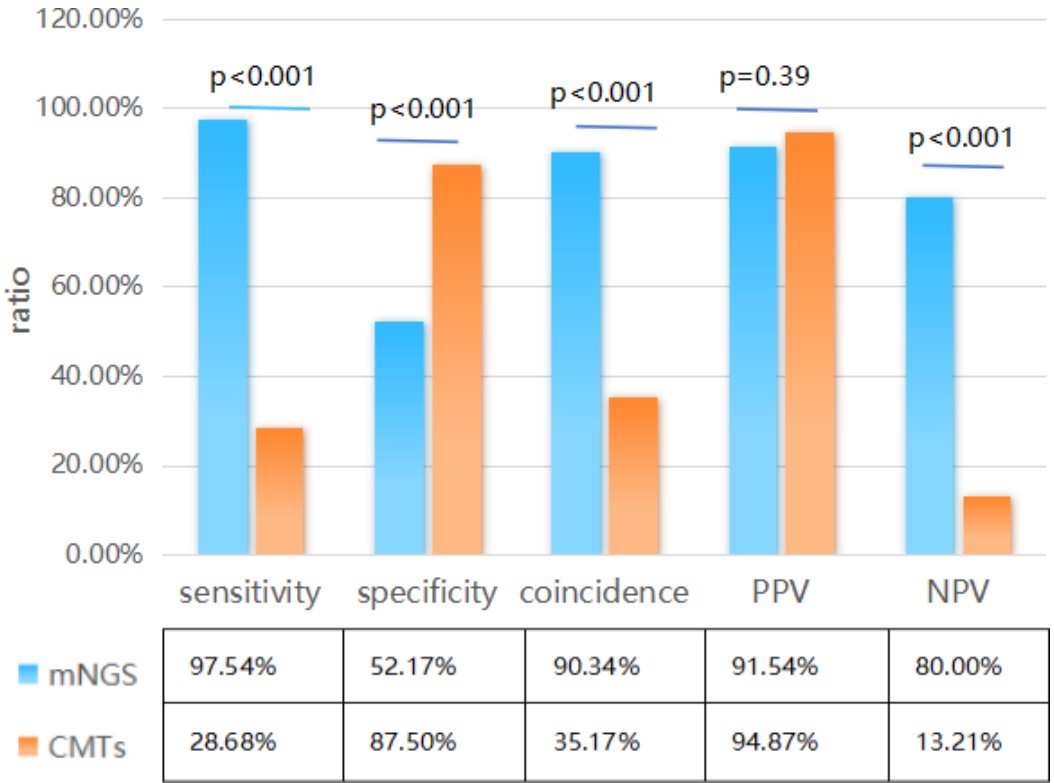

**FIG 3** The comparison of sensitivity, specificity, coincidence, positive predictive value, and negative predictive value between mNGS and CMTs.

treated with vancomycin and linezolid due to the detection of *S. pneumoniae* but died due to a longer course of disease and older age combined with cerebral infarction. One case died from poor cardiopulmonary function caused by *E. faecium*, *yeast*, and *Herpes virus* coinfection.

## DISCUSSION

For patients with severe pneumonia, because of the long clinical course and the empirical use of antibiotics, it become increasingly difficult to identify the etiology of pneumonia, resulting in the failure of therapy and excessive use of antibiotics. Both sputum and BALF are widely used in the diagnosis of RTIs, while sputum is contaminated with pharyngeal microorganism, and it has low sensitivity with CMTs and is susceptible to empirical antibiotic treatments (28), while BALF is rarely influenced by pharyngeal contamination and performs a higher accuracy (29). However, empirical antibiotics treatment will be usually unavoidable once pneumonia patients are admitted to the hospital. mNGS can sequence all nucleic acids simultaneously in a sample independent of the priori selection of targeted pathogens, improving the sensitivity of pathogen detection and having a revolutionary impact on microbiological diagnosis. Based on the number and relative abundance of reads, mNGS can be used for semi-quantitative detection and improving precise treatment (30). Compared with CMTs, mNGS showed

**TABLE 2** Diagnostic performance of mNGS and CMTs in respiratory tract infections[a]

| Workflow | No. of samples categorized as | | | | Sensitivity (%) | Specificity (%) | Coincidence (%) | PPV (%) | NPV (%) |
|---|---|---|---|---|---|---|---|---|---|
| | TP | FP | TN | FN | | | | | |
| mNGS | 119 | 11 | 12 | 3 | 97.54 | 52.17 | 90.34 | 91.54 | 80 |
| CMTs | 37 | 2 | 14 | 92 | 28.68 | 87.5 | 35.17 | 94.87 | 13.21 |

[a]mNGS, metagenomic next-generation sequencing; CMTs, conventional microbiological tests; PPV, positive predictive value; NPV, negative predictive value; TP, true positive; FP, false positive; TN, true negative; FN, false negative.

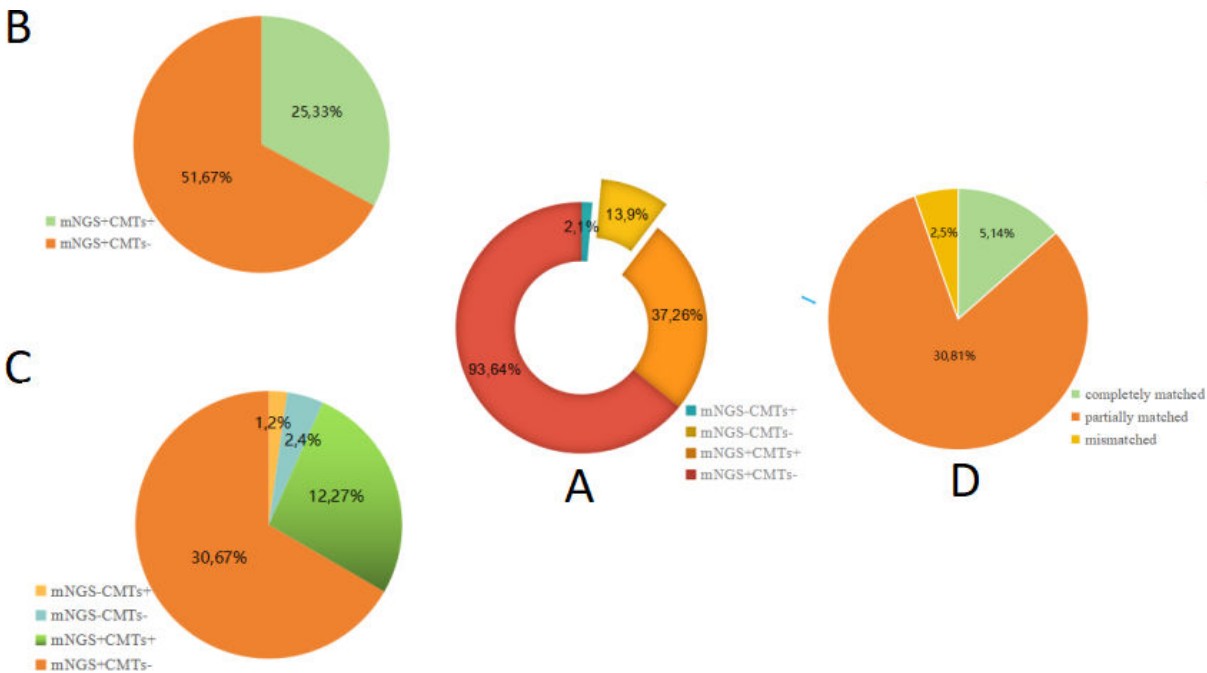

**FIG 4** Concordance of diagnosis between mNGS and CMTs. (A) Ninety-three cases (64.14%, 93/145) were both positive, 13 cases (8.97%, 13/145) were negative for both methods, and 2 cases (1.38%, 2/145) were positive by CMTs only. (B) In the mixed infections group, 25 cases (32.89%, 25/76) were both positive and 51 cases (67.11%, 51/76) were positive by mNGS only. (C) In the monobacterial infections group, 12 cases (21.82%, 12/55) were both positive and 2 cases (3.64%, 2/55) were both negative. (D) Among the 37 double-positive cases, 30 cases (81.08%, 30/37) were matched, 5 cases (13.51%, 5/37) were partially matched, and 2 cases (5.41, 2/37) were completely mismatched.

a significant advantage in detecting the causative pathogens, which had a higher positive rate and a wider pathogen spectrum, and was less affected by the empirical antibiotic usage, which was consistent with previous studies (31, 32). The results indicate that mNGS can be used as a supplementary diagnostic method for severe pneumonia, especially for patients who were treated with antibiotics before sample collection.

mNGS has potential advantages in terms of sensitivity for detecting lung diseases (33). This study showed that the sensitivity and coincidence rate of mNGS had a more significant advantage than those of CMTs (97.54% vs 28.68%, $P < 0.001$; 90.34% vs 35.17%, $P < 0.001$), which was consistent with a previous study (34). Different from a previous study (35), the specific rate of mNGS was lower than that of CMTs, which may be due to the following reasons: (i) CMTs showed more false-negative results caused by fastidious microorganisms and by the previous initiated antibiotics treatment before the collection of BALF (36); (ii) BALF derived from the respiratory tract may be contaminated with colonizers, and mNGS has higher sensitivity, contributing to its relatively higher false-positive results; and (iii) the enrolled patients are different from other studies. In this study, 37 cases were positive for both methods, only 5 cases were completely matched, 2 cases were completely mismatched, and the remaining 30 cases were partially matched, which may be due to the low positive rate and narrow detection range of CMTs, while mNGS shows higher sensitivity and a wide-range pathogen coverage.

mNGS shows a higher positive rate to multiple pathogens, including bacteria, fungi, virus, and other microbes (such as *M. tuberculosis*, *Non-tuberculous mycobacterium*, and other atypical pathogens). RTIs can be infected by multiple pathogens simultaneously. Atypical pathogens such as *Spirochaeta*, *Mycoplasma*, and *Legionella* are more likely to be ignored. mNGS greatly improved the proportion of pneumonia caused by atypical pathogens (37, 38). We detected 14 *Aspergillus*, 4 *C. psittaci*, 3 *P. jirovecii*, 3 *M. tuberculosis*, 7 *Actinomyces*, and many other atypical pathogens. Our results showed that *S. pneumoniae*, *S. constellatus*, *P. micra*, *A. baumannii*, *C. albicans*, and *HSV1* were

common pathogens of severe pneumonia. *S. pneumoniae* was considered as the most common bacterial cause of pneumonia in all patients of all ages (39, 40) and had the largest proportion (22.90%, 30/131) in all enrolled patients in this study, indicating its high pathogenicity. Thus, early diagnosis and timely treatment are indispensable to improve the prognosis of pneumonia patients. It should be noted that, similar to the previous study (41), only 5.34% (7/131) of patients was diagnosed with *S. pneumoniae* by culture in this study, which was significantly lower than that by mNGS. Therefore, mNGS, as a superior diagnostic method, contributes to improved clinical identification of *S. pneumoniae* infection. Mixed infections were commonly accompanied by several pathogens in clinical practice, mixed infections accounted for 58.02% in this study, and the most common type was bacterial-viral coinfection. In addition, the most complicated mixed infections were bacterial-fungal-viral-atypical pathogen coinfection, which had 5 cases. The positive rate of mNGS in mixed infections was 100%, indicating that mNGS had a significant advantage in mixed infections and etiological diagnosis, consistent with the results of previous studies (36, 42), which also suggested that doctors should inspect mNGS timely when they suspected atypical pathogen infection.

Most fungi were non-pathogenic in healthy individuals; however, patients with cancer or other immunocompromised conditions have higher infection risk, which always cause mixed infections together with virus, because viral infection can damage the mucosal linings of respiratory tracts (43). Multiple factors, including epidemiology, clinical manifestations, laboratory test results, imaging results, conventional diagnostic results, and outcomes after previous anti-infective treatments, were considered to help distinguish the pathogen. Studies showed that one kind of microorganism will indirectly increase the risk of colonization for other microorganisms (44, 45). Although it is difficult to extract DNA of *Aspergillus* because of its polysaccharide cell wall, however, mNGS still showed a higher detection rate than CMTs in this study. Literature indicated that fungal and viral infection increased the risk of bacterial infection (44, 45). Xie et al. (46) detected more *Chlamydia psittaci* and *Chlamydia abortus* coinfection by applying mNGS in BALF with RTI patients and showed that pneumonia caused by *Chlamydia psittaci* without timely treatment would have a high mortality rate, which needs to be determined by more specific tests, due to the lack of typical epidemiological history, clinical manifestations, and imaging examination. The culture of *M. tuberculosis* is a long cycle, and most *N. mycobacteria* is difficult to culture (47). *Pneumocystis yersinia* is a kind of ascomycetes, is an important pathogen of severe pneumonia in immunosuppressed patients, and mainly causes diffuse alveolar injury in alveolar epithelium, which also causes mixed infections with other pathogens easily. Three *P. yersinia* in this study were detected from immunodeficient patients, which was consistent with previous studies (48, 49). Some studies showed that the *P. yersinia* reads are usually higher than those of other intracellular pathogens in the same BALF (50, 51). It is difficult to be distinguished from pulmonary GVHD, pulmonary infiltration of connective tissue disease, and viral pneumonia. Traditional methods have the limitations of low positive and high false-positive rates, whereas novel molecular diagnostics targeting novel gene targets, e.g., the mitochondrial small subunit rRNA gene, may provide a more sensitive and specific testing performance (52).

In this study, 131 cases were diagnosed based on mNGS results. The antibiotics for 39 patients were adjusted according to the results of mNGS combined with clinical characteristics. Out of the 55 monobacterial infections, 14 cases had adjusted antibiotics or increased corresponding antibiotics due to the detection of bacteria, *C. psitsiti*, *Ureaplasma parvum*, or *M. tuberculosis*. The condition of 11 cases improved greatly after precise treatment timely. Among the 76 mixed infections, 12 cases had increased antifungal drugs due to the detection of fungi by mNGS. This study shows that mNGS has a greater advantage than CMTs in fungal detection, consistent with previous studies (48, 49). It is difficult to distinguish whether the fungi is colonization or infection, which requires the clinician to make a comprehensive judgment.

There are certain limitations in this study. First, mNGS can only detect pathogens; it is difficult to determine if the microbe is dead or alive. Atypical pathogens failed to be further verified by traditional criteria methods or real-time PCR detection. Second, this study did not statistically compare the viral detection rate between mNGS and traditional viral detection. Third, the abundance of mNGS is only quantitative and cannot fully represent the proportion of pathogens, and a series of studies are needed to establish a uniform standard for pathogen quantification. Meanwhile, the accurate interpretation of mNGS should be combined with clinical manifestations and conventional laboratory-based diagnosis.

In conclusion, mNGS has a significant advantage in the diagnosis of RTIs in critically ill patients, especially for patients who have used antibiotics before admission or showed complicated clinical symptoms. Detection of pathogens timely and comprehensively can guide clinical precise usage of antibiotics, improving patient prognosis and reducing mortality.

## MATERIALS AND METHODS

### Patients and study design

This retrospective study consecutively enrolled 145 patients with suspected severe pneumonia who were admitted to the respiratory and care medicine departments of the Affiliated Hospital of Jining Medical University between June 2021 and June 2022. The enrollment criteria were as follows: (i) patients have suspected severe pneumonia; (ii) BALF samples were qualified, and mNGS and CMTs were performed simultaneously; (iii) the clinical data were complete; and (iv) age was ≥18.

The patient's family signed written informed consent, and BALF samples were extracted by the standard operation and detected by CMTs and mNGS simultaneously. All the tests were performed by Medical Laboratory Science, Affiliated Hospital of Jining Medical University. CMTs using BALF were applied as part of ordered testing, including Gram stain and quantitative aerobic bacterial culture, fungal culture, galactomannan antigen testing, and mycobacterial stain. Except Gram stain, not all testing was performed on each specimen, while Gram stain was performed on each specimen. Enrolled samples were frozen at −80℃. The clinical data of all enrolled patients were collected, which included gender, age, underlying disease, laboratory test results, antibiotic treatment, length of hospital stay, and the survival status of patients for 3 months that was followed up by telephone. According to the type of infections, they were classified into monobacterial infections group and mixed infections group. The mixed infections group is infected with two or more types of microorganisms.

### Sample treatment

A total of 3 mL BALF was collected and inactivated in a water bath at 65℃ for 30 min. A new 1.5-mL centrifuge tube containing 7.2 μL lysozyme, 7.5 μL DTT solution, and 450 μL inactivated sample was kept at 30℃ for 10 min. A spiral tube containing the above sample and 250-μL glass beads were placed in a Labnet VX-200 oscillator which was set for a maximum oscillation of 20 min. Three hundred microliters of supernatant was collected in a 1.5-mL centrifuge tube for later use after centrifugation.

### Extraction of DNA nucleic acid

DNA was extracted strictly in accordance with the instructions of the DNA purification kit (Huada Biotechnology Co. LTD., Wuhan, China). DNA concentration was determined by dsDNA HS Assay Kit 4.0. Appropriate concentration was used to construct DNA library by DNA fragmentation, end repair, adapter connection, and PCR. DNA fragmentation was performed using the DNA enzyme digestion reaction kit (Bada Biotechnology Co. LTD.) (53). PCR reaction conditions were as follows: 98℃ for 2 min, followed by 12 cycles of 98℃ for 15 s, 56℃ for 15 s, and 72℃ for 30 s, with a final extension at 72℃ for 5 min.

## Construction of DNA libraries

DNA libraries and sequencing libraries were prepared using the PMse high-throughput DNA assay kit (Huada Biotechnology Co. LTD.). A Qubit dsDNA HS Assay Kit 4.0 fluorometer was used to measure the DNA concentration in each sample. Qualified libraries were pooled in 0.2-mL PCR tubes and then circularized to form a single-chain circular structure. DNA nanospheres were prepared using the MiSeqDx Kit (Huada Biotechnology Co. LTD.) and loaded into the sequencing chip and sequenced using the Huada mNGS platform (54). Negative and positive quality controls were participated in the whole test process for quality control and contamination reduction.

## Bioinformatic analysis

The effective sequencing data volume should not be less than 20 M. The qualified data were further filtered by bioinformatics analysis to remove low-quality sequences and short sequences (<35 bps) using the FASTQ format which included low-quality read filtering, low-complexity read filtering, and adapter trimming. Subsequently, Burrows Wheeler Alignment (version 0.7.10) was performed by mapping the filtered sequences to a human reference database to identify and exclude human host sequences. The remaining data were aligned to the pathogen metagenomics database, which cover 20,364 pathogen genomes, including 11,910 bacterial genomes, 1,046 fungi associated with human infection, 7,103 viral taxa complete genome sequences, and 305 parasites associated with human infection (55). All parameters of detected pathogens were classified and recorded, including read counts, relative abundance, genome coverage, and depth.

Infectious pathogens are considered positive if they meet any conditions of mNGS as follows, including bacteria, virus, and fungi: (i) there is >30% relative abundance at the genus level in bacteria (except *M. tuberculosis*) or fungi; (ii) a single species of bacteria or fungi has at least 50 unique reads; (iii) *M. tuberculosis* is considered positive even if there's only one unique read, due to the difficulty of DNA extraction and low possibility for contamination; (iv) the strict sequence number of the virus is not less than 3; and (v) if there are relevant clinical conditions, the pathogen can be diagnosed even if the sequence number is less than 50 (25, 56).

The results of mNGS and CMTs was evaluated against the final clinical diagnoses by two experienced clinicians based on multiple clinical factors, including the epidemiology, clinical manifestations, laboratory examinations, imaging findings, therapeutic outcomes. When the final diagnosis was consistent with the results of CMTs or mNGS, it was defined as coincidence, including true positive (TP) or true negative (TN), otherwise it was defined as false positive (FP) or false negative (FN).

## Statistical analysis

Continuous variables were expressed as median [first quartile ($Q_1$), third quartile ($Q_3$)]. Categorical variables were expressed as numbers and percentages. All statistics were reported as absolute values and determined using the *Wilson's* method, $\chi^2$ test, and *Mann-Whitney U*. All statistical analyses were conducted in the Beckman Coulter *Dx*AI platform (https: //www. xsmartanalysis. com/beckman/login/). A two-tailed *P* value of 0.05 was considered statistically significant. This study was approved by the Institutional Medical Ethics Committee of our hospital (No. 2024-02-C007).

### ACKNOWLEDGMENTS

We are grateful to all the participants. Min Kong and Liqing Jiang contributed to the design and execution of the study and writing of the manuscript. Xiaowei Liu, Hong Chang, Haixin Dong, Jinyan Yan, and Xiaoya Zhou contributed to the raw sequence data upload process and their assistance with the project. We also deeply appreciate the doctors'assistance with sample collection.

This work was supported by the Research Fund for Academician Lin He New Medicine (No. JYHL2022MS05) and Jining City Key Research and Development Plan Project (No. 2021YXNS109 and No. 2023YXNS178).

## AUTHOR AFFILIATIONS

[1]Medical Laboratory Science, Affiliated Hospital of Jining Medical University, Jining Medical University, Shandong Jining, China
[2]Department of Intensive Care Unit, Affiliated Hospital of Jining Medical University, Jining Medical University, Shandong Jining, China
[3]Medical Laboratory of Jining Medical University, Lin He's Academician Workstation of New Medicine and Clinical Translation in Jining Medical University, Jining Medical University, Shandong Jining, China

## AUTHOR ORCIDs

Xiaohang Hu http://orcid.org/0009-0003-0337-4841
Min Kong http://orcid.org/0009-0003-0210-4399

## AUTHOR CONTRIBUTIONS

Xiaohang Hu, Conceptualization, Data curation, Formal analysis, Funding acquisition, Methodology, Project administration, Resources, Writing – original draft, Writing – review and editing | Liqing Jiang, Conceptualization, Formal analysis, Methodology, Software, Validation | Xiaowei Liu, Data curation, Investigation, Resources, Software, Validation | Hong Chang, Conceptualization, Data curation, Investigation, Supervision, Validation, Visualization | Haixin Dong, Data curation, Investigation, Methodology, Project administration, Resources, Supervision, Validation, Visualization | Jinyan Yan, Data curation, Formal analysis, Investigation, Project administration, Software, Supervision, Visualization | Xiaoya Zhou, Methodology, Project administration, Validation, Visualization.

## ADDITIONAL FILES

The following material is available online.

Open Peer Review

**PEER REVIEW HISTORY (review-history.pdf).** An accounting of the reviewer comments and feedback.

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
