## [Reviewer comments · Microbiology Spectrum]

Microbiology Spectrum

The Diagnostic Value of Bronchoalveolar Lavage Fluid Metagenomic Next-generation Sequencing in Critically Ill Patients with Respiratory Tract Infections

Xiaohang Hu, Liqing Jiang, Xiaowei Liu, hong Chang, Haixin Dong, Jinyan Yan, Xiaoya Zhou, and Min Kong

Corresponding Author(s): Min Kong, Jining Medical University

Review Timeline:

Submission Date:	March 9, 2024
Editorial Decision:	April 18, 2024
Revision Received:	May 11, 2024
Accepted:	May 18, 2024

Editor: Benjamin Liu

Reviewer(s): Disclosure of reviewer identity is with reference to reviewer comments included in decision letter(s). The following individuals involved in review of your submission have agreed to reveal their identity: Kaan Çeylan (Reviewer #3)

Transaction Report:

DOI: <https://doi.org/10.1128/spectrum.00458-24>

Re: Spectrum00458-24 (The diagnostic value of bronchoalveolar lavage fluid metagenomic next-generation sequencing in critically ill patients with respiratory tract infections)

Dear Dr. Min Kong:

Thank you for the privilege of reviewing your work. Below you will find my comments, instructions from the Spectrum editorial office, and the reviewer comments.

Editor's suggestions:

1. Line 51 . "The sensitivity and correspondence of mNGS were". "correspondence" is confusing and should be changed.
2. *Pneumocystis yerinii* should be *Pneumocystis jirovecii*
3. Line 70-71: "Respiratory tract infections (RTIs) is one of the important causes of death worldwide, with high morbidity and mortality." There are no references to support this statement. More references should be cited, with the following four as examples to be cited.

Role of Host Immune and Inflammatory Responses in COVID-19 Cases with Underlying Primary Immunodeficiency: A Review. *J Interferon Cytokine Res.* 2020 Dec;40(12):549-554. doi: 10.1089/jir.2020.0210. PMID: 33337932; PMCID: PMC7757688.

Clinical significance of measuring serum cytokine levels as inflammatory biomarkers in adult and pediatric COVID-19 cases: A review. *Cytokine.* 2021 Jun;142:155478. doi: 10.1016/j.cyto.2021.155478. Epub 2021 Feb 23. PMID: 33667962; PMCID: PMC7901304.

The Brief Case: Ventilator-Associated *Corynebacterium accolens* Pneumonia in a Patient with Respiratory Failure Due to COVID-19. *J Clin Microbiol.* 2021 Aug 18;59(9):e0013721. doi: 10.1128/JCM.00137-21. Epub 2021 Aug 18. PMID: 34406882; PMCID: PMC8372998.

Influenza A virus directly modulates mouse eosinophil responses. *J Leukoc Biol.* 2020 Jul;108(1):151-168. doi: 10.1002/JLB.4MA0320-343R. Epub 2020 May 9. PMID: 32386457; PMCID: PMC7859173.

4. Line 75-76: "Bacteria, fungi, viruses and some atypical pathogens can all cause respiratory tract infections". More references should be cited, with the following three as examples to be cited.

Universal PCR Primers Are Critical for Direct Sequencing-Based Enterovirus Genotyping. *J Clin Microbiol.* 2016 Dec 28;55(1):339-340. doi: 10.1128/JCM.01801-16. PMID: 28031445; PMCID: PMC5228251.

Optimization and evaluation of a novel real-time RT-PCR test for detection of parechovirus in cerebrospinal fluid. *J Virol Methods.* 2019 Oct;272:113690. doi: 10.1016/j.jviromet.2019.113690. Epub 2019 Jul 5. PMID: 31283959.

5. Line 81-84: "However, traditional testing are widely applied in clinical settings such as CMTs, immunological tests and polymerase chain reaction (PCR), but they are poor timeliness, low pathogen coverage rate and low positive rate, which is difficult to meet the diagnostic needs of critically ill patients". This statement needs to be changed to "However, traditional testing are widely applied in clinical settings such as CMTs, and immunological tests, but they are poor timeliness, low pathogen coverage rate and low positive rate, which is difficult to meet the diagnostic needs of critically ill patients". More references should be cited, with the following one as an example to be cited.

The Brief Case: *Capnocytophaga sputigena* Bacteremia in a 94-Year-Old Male with Type 2 Diabetes Mellitus, Pancytopenia, and Bronchopneumonia. *J Clin Microbiol.* 2021 Jun 18;59(7):e0247220. doi: 10.1128/JCM.02472-20. Epub 2021 Jun 18. PMID: 34142857; PMCID: PMC8218739.

6. Line 93-94: statement "Metagenomic next-generation sequencing(mNGS) is a high-throughput sequencing technology, which has the advantages of unculture, unbiased, unhypothesis, wide coverage, et al." should be changed to "Metagenomic next-generation sequencing(mNGS) is a high-throughput sequencing technology, which has the advantages of unculture, unbiased, unhypothesis, wide coverage, and so on." More references should be cited, with the following one as an example to be cited.

Laboratory diagnosis of CNS infections in children due to emerging and re-emerging neurotropic viruses. *Pediatr Res.* 2024 Jan;95(2):543-550. doi: 10.1038/s41390-023-02930-6. Epub 2023 Dec 2. PMID: 38042947.

7. Please move acknowledgement after discussion.

8. Line 394-395: statement: "Traditional methods have the limitations of low positive and high false positive rate." should be

changed to "Traditional methods have the limitations of low positive and high false positive rate whereas novel molecular diagnostics targeting novel gene targets, e.g., the mitochondrial small subunit rRNA gene, may provide a more sensitive and specific testing performance". More references should be cited, with the following one as an example to be cited.

Development and Evaluation of a Fully Automated Molecular Assay Targeting the Mitochondrial Small Subunit rRNA Gene for the Detection of *Pneumocystis jirovecii* in Bronchoalveolar Lavage Fluid Specimens. *J Mol Diagn.* 2020 Dec;22(12):1482-1493. doi: 10.1016/j.jmoldx.2020.10.003. Epub 2020 Oct 15. Erratum in: *J Mol Diagn.* 2021 Apr;23(4):506. PMID: 33069878.

Please return the manuscript within 30 days; if you cannot complete the modification within this time period, please contact me. If you do not wish to modify the manuscript and prefer to submit it to another journal, notify me immediately so that the manuscript may be formally withdrawn from consideration by Spectrum.

Revision Guidelines

Sincerely,
Benjamin Liu
Editor
Microbiology Spectrum

Reviewer #2 (Comments for the Author):

Despite its advantages in detecting pathogens, mNGS technology requires extensive evaluations of accuracy, precision, reportable range, and reference range to ensure reliable results.

The test's low specificity is a significant concern in diagnosing pulmonary infections with BAL specimens. It is crucial to differentiate between commensals/colonizers and pathogens using appropriate methods or techniques. Can you confirm whether the authors utilized any such methods or techniques?

It is concerning that the authors did not provide information on the conventional microbiology methodology used to detect pathogens or specify the colony count considered as significant growth of bacteria in BAL specimens. Such crucial details are necessary for the comparison of mNGS with conventional methods. Therefore, it is highly recommended that the authors provide this important data.

Actinomyces are colonizers, but their significance varies. I would like to know the species of the isolated Actinomyces.

What percentage of BAL fluid specimens undergo direct microscopy methods such as Gram stain and/or KOH mount?

Including a table that compares culture results with mNGS would significantly enhance the clarity of the presented data. It would provide a clearer picture of the findings and help researchers to draw more accurate conclusions. Therefore, it is highly recommended that authors include such a table in their research paper.

What is the gold standard test used to find the sensitivity, specificity, PPV, and NPV of mNGS?

Line 283 and 284 indicate that mNGS has detected the pathogen *Fumigus*. Can you please provide me with more information about this pathogen?

It's crucial to follow the journal's guidelines when preparing a research paper. The acknowledgment section is in between the methods and results section. Correct the error.

Please ensure that the font size and format are adjusted in accordance with the guidelines of the journal. Also, kindly check for any spelling, grammar, or punctuation errors that may need to be corrected.

Reviewer #3 (Comments for the Author):

Dear author,

The diagnostic value of bronchoalveolar lavage fluid metagenomic next-generation sequencing in critically ill patients with respiratory tract infections has been reviewed. The study needs to be reviewed on some issues. These considerations are stated below.

1. It is necessary to specify the conventional methods in more detail in the materials and methods section.
2. The expression "fliud" in line 48 should be corrected to "fluid".
3. "Streptococcus pneumoniae was the most common bacterial pathogen." the statement should be more descriptive.
4. The abstract part does not summarize the study literally. It is expected that a person who reads the abstract part will have a rough knowledge about the construction and results of the study.
5. The expression "classified" on line 154 should be changed to "classified".
6. The expression "basical" on line 189 should be changed to "basically" or "basic".
7. It is stated that the study is retrospective in line 110 and prospective in line 189, and the same expression should be used everywhere so as not to cause confusion.
8. It should be explained how the patient who is identified as a false positive by the conventional method is evaluated as a false positive.
9. The expression "tigacycline" on line 315 should be changed to "tigecycline".
10. The expression "linzolid" on line 331 should be changed to "linezolid".
11. There are serious grammatical errors in the text and they need to be corrected.

I wish good work

Thank you for your letter and thanks for your careful review to our manuscript entitled "The diagnostic value of metagenomic next-generation sequencing using bronchoalveolar lavage fluid in critically ill patients with respiratory tract infections". Those comments are all valuable and very helpful for revising and improving our paper, as well as the important guiding significance to our researches. We have studied comments carefully and have made correction which we hope meet with approval. Revised portion are marked in red in the paper. The main corrections in the paper and the responds to the reviewer's comments are as follows:

Responds to the reviewer's comments:

Reviewer #1:

Comments 1: Line 51. "The sensitivity and correspondence of mNGS were". "correspondence" is confusing and should be changed.

Answer: The word "correspondence" had been changed to "coincidence" in line 30.

Comments 2: *Pneumocystis yerinii* should be *Pneumocystis jirovecii*.

Answer: All the words "*Pneumocystis yerinii*" in our paper had been changed to "*Pneumocystis jirovecii*" in line 66 and 310.

Comments 3: Line 70-71: "Respiratory tract infections (RTIs)is one of the important causes of death worldwide, with high morbidity and mortality." There are no references to support this statement. More references should be cited, with the following four as examples to be cited.

Answer: The statement "Respiratory tract infections (RTIs) is one of the important causes of death worldwide, with high morbidity and mortality." had been changed to

"Of all infectious disease categories, respiratory tract infections (RTIs) impart the greatest mortality both worldwide and the United States, which are extremely frequent in both adults and children, representing an increased economic burden, morbidity and mortality." in line 53-56. Some references had also been cited.

Comments 4: Line 75-76: "Bacteria, fungi, viruses and some atypical pathogens can all cause respiratory tract infections". More references should be cited, with the following three as examples to be cited.

Answer: The statement "Bacteria, fungi, viruses and some atypical pathogens can all cause respiratory tract infections" had been changed to "Various pathogens, including bacteria, fungi, virus, parasitic and some atypical pathogens had been reported to be aetiology of RTIs (4 to 9). With the report and recognition of coronavirus disease (COVID-19) in December 2019 in the city of Wuhan in Hubei province, the role of respiratory viruses had been becoming more dominant (10). Liu BM et al (11) found that primary immunodeficiency patients may be susceptible to SARS-CoV-2 infection after literature review." in line 60-65 . Some references had also been cited.

Comments 5: Line 81-84: "However, traditional testing are widely applied in clinical settings such as CMTs, immunological tests and polymerase chain reaction (PCR), but they are poor timeliness, low pathogen coverage rate and low positive rate, which is difficult to meet the diagnostic needs of critically ill patients". This statement needs to be changed to "However, traditional testing are widely applied in clinical settings such as CMTs, and immunological tests, but they are poor timeliness, low pathogen coverage rate and low positive rate, which is difficult to meet the diagnostic needs of

critically ill patients". More references should be cited, with the following one as an example to be cited.

Answer: The statement "However, traditional testing are widely applied in clinical settings such as CMTs, immunological tests and polymerase chain reaction (PCR), but they are poor timeliness, low pathogen coverage rate and low positive rate, which is difficult to meet the diagnostic needs of critically ill patients" had been changed to "Currently, traditional etiological detection methods are commonly applied for the diagnosis of RTIs such as CMTs and immunological tests, but they are poor timeliness, low pathogen coverage and positive detection rate, which is difficult to meet the diagnostic needs of critically ill patients (13)." in line 71-75. Some references had been cited.

Comments 6: Line 93-94: statement "Metagenomic next-generation sequencing (mNGS) is a high-throughput sequencing technology, which has the advantages of unculture, unbiased, hypothesis-free, wide coverage, et al." should be changed to "Metagenomic next-generation sequencing(mNGS) is a high-throughput sequencing technology, which has the advantages of unculture, unbiased, hypothesis-free, wide coverage, and so on." More references should be cited, with the following one as an example to be cited.

Answer: The statement "Metagenomic next-generation sequencing (mNGS) is a high-throughput sequencing technology, which has the advantages of unculture, unbiased, hypothesis-free, wide coverage, et al." had been changed to "Metagenomic next-generation sequencing (mNGS) is a high-throughput sequencing technology,

which has the advantages of unculture, unbiased, un hypothesis, wide coverage, and so on (21)." in line 89-91. Some references had been cited.

Comments 7: Please move acknowledgement after discussion.

Answer: I had move acknowledgement after material and methods following the journal's guidelines in line 463-472.

Comments 8: Line 394-395: statement: "Traditional methods have the limitations of low positive and high false positive rate." should be changed to "Traditional methods have the limitations of low positive and high false positive rate whereas novel molecular diagnostics targeting novel gene targets, e.g., the mitochondrial small subunit rRNA gene, may provide a more sensitive and specific testing performance". More references should be cited, with the following one as an example to be cited.

Answer: The statement: "Traditional methods have the limitations of low positive and high false positive rate." had been changed to "Traditional methods have the limitations of low positive and high false positive rate, whereas novel molecular diagnostics targeting novel gene targets, e.g., the mitochondrial small subunit rRNA gene, may provide a more sensitive and specific testing performance (52)." in line 355-359. Some references had been cited.

Reviewer #2

Comments 1: Despite its advantages in detecting pathogens, mNGS technology requires extensive evaluations of accuracy, precision, reportable range, and reference range to ensure reliable results.

Answer: our worker detected the samples provided by the manufacturer containing some known pathogens to verify the accuracy and precision of the method after installation, but did not carried out detection of different gradient concentrations. The reportable range and reference range were provided by the manufacturer.

Comments 2: The test's low specificity is a significant concern in diagnosing pulmonary infections with BAL specimens. It is crucial to differentiate between commensals/colonizers and pathogens using appropriate methods or techniques. Can you confirm whether the authors utilized any such methods or techniques?

Answer: Currently, we have no appropriate and credible methods to differentiate colonizers and pathogens. We are also very confused. In the future, we will also conduct relevant research on this aspect of confusion.

Comments 3: It is concerning that the authors did not provide information on the conventional microbiology methodology used to detect pathogens or specify the colony count considered as significant growth of bacteria in BAL specimens. Such crucial details are necessary for the comparison of mNGS with conventional methods. Therefore, it is highly recommended that the authors provide this important data.

Answer: We inoculated 10 μ L of BALF. Cultures were examined at 24 h and 48 h, and predominant organisms were identified when there was only one type of bacteria

or the quantity of one bacterial type is larger than others on semiquantitative culture.

Comments 4: Actinomyces are colonizers, but their significance varies. I would like to know the species of the isolated Actinomyces.

Answer: The *Actinomycetes* isolated in this study included 3 *Actinomyces israelii*, 2 *Actinomyces naeslundii*, 2 *Actinomycesodontolyticus* and 1 *Actinomycetes Graveniaae*.

Comments 5:What percentage of BAL fluid specimens undergo direct microscopy methods such as Gram stain and/or KOH mount?

Answer: All the BALF specimens performed Gram stain before detection.

Comments 6: Including a table that compares culture results with mNGS would significantly enhance the clarity of the presented data. It would provide a clearer picture of the findings and help researchers to draw more accurate conclusions. Therefore, it is highly recommended that authors include such a table in their research paper.

Answer: A table including some indexes which were used to compare mNGS and CMTs is added in this manuscripts, it is table 2 in line 210-216.

Comments 7: What is the gold standard test used to find the sensitivity, specificity, PPV, and NPV of mNGS?

Answer: Firstly, The final performance of mNGS and CMTs were evaluated against the final clinical diagnosis by two experienced clinicians based on multiple, including the epidemiology, clinical manifestations, laboratory examinations, imaging findings, therapeutic effect observation. When the microorganisms detected by CMTs or mNGS was consistent with final diagnosis, it is defined as coincidence, including true

positive (TP) or true negative (TP), whereas it is defined as false positive (FP) or false negative (FN). Then, we calculate Sensitivity, Specificity, coincidence, PPA and NPA according to the correlated formulas.

Comments 8: Line 283 and 284 indicate that mNGS has detected the pathogen Fumigus. Can you please provide me with more information about this pathogen?

Answer: I am sorry for the misrepresentation, in fact I want to express *Aspergillus*, we detect 8 *Aspergillus fumigatus*, 4 *Aspergillus fumigatus*, 2 *Aspergillus niger* in this study. The statement has been changed to “mNGS detected some pathogens that were difficult to be detected by CMTs, including *Aspergillus*, *Rhizopus*, *Haemophilus*, *M. tuberculosis*, *Mycoplasma*, various virus and so on.” in line 207-209.

Comments 9: It's crucial to follow the journal's guidelines when preparing a research paper. The acknowledgment section is in between the methods and results section. Correct the error.

Answer: I had move material and methods after discussion and move acknowledgement after material and methods following the journal's guidelines.

Comments 10: Please ensure that the font size and format are adjusted in accordance with the guidelines of the journal. Also, kindly check for any spelling, grammar, or punctuation errors that may need to be corrected.

Answer: I had correct spelling, grammar, punctuation errors, the font size and format in accordance with the guidelines of the journal.

Reviewer #3 :

Comments 1: It is necessary to specify the conventional methods in more detail in the materials and methods section.

Answer: The conventional methods had been added in the materials and methods section in line 394-396.

Comments 2: The expression "fliud" in line 48 should be corrected to "fluid".

Answer: The expression "fliud" had been corrected to "fluid" in line 28 and 50.

Comments 3: "Streptococcus pneumoniae was the most common bacterial pathogen." the statement should be more descriptive.

Answer: For the statement "Streptococcus pneumoniae was the most common bacterial pathogen.", The more descriptive had been added in dicussion section in line 313-315.

Comments 4: The abstract part does not summarize the study literally. It is expected that a person who reads the abstract part will have a rough knowledge about the construction and results of the study.

Answer: All the abstract part had been modified in line 23-43.

Comments 5: The expression "categorized" on line 154 should be changed to "classified".

Answer: The expression "categorized" had been changed to "classified" in line 402.

Comments 6: The expression "basical" on line 189 should be changed to "basically" or "basic".

Answer: The expression "basical" had been changed to "basic" in line 112.

Comments 7: It is stated that the study is retrospective in line 110 and prospective in line 189, and the same expression should be used everywhere so as not to cause confusion.

Answer: The two expressions had been changed to "retrospective" in line 106 and 385.

Comments 8: It should be explained how the patient who is identified as a false positive by the conventional method is evaluated as a false positive.

Answer: The statement "The final performance of mNGS and CMTs was evaluated against the final clinical diagnoses by two experienced clinicians based on multiple, including the epidemiology, clinical manifestations, laboratory examinations, imaging findings, therapeutic effect observation. When the microorganisms detected by CMTs or mNGS was consistent with final diagnosis, it is defined as coincidence, including true positive (TP) or true negative (TN), whereas it is defined as false positive (FP) or false negative (FN)." had been added in line 448-454.

Comments 9: The expression "tigacycline" on line 315 should be changed to "tigecycline".

Answer: The expression "tigacycline" had been changed to "tigecycline" in line 247.

Comments 10: The expression "linzolid" on line 331 should be changed to "linezolid".

Answer: The expression "linzolid" had been changed to "linezolid" in line 265.

Comments 11: There are serious grammatical errors in the text and they need to be

corrected.

Answer: I had correct spelling, grammar, punctuation errors, the font size and format in accordance with the guidelines of the journal.

Re: Spectrum00458-24R1 (The Diagnostic Value of Bronchoalveolar Lavage Fluid Metagenomic Next-generation Sequencing in Critically Ill Patients with Respiratory Tract Infections)

Dear Dr. Min Kong:

Your manuscript has been accepted, and I am forwarding it to the ASM production staff for publication. Your paper will first be checked to make sure all elements meet the technical requirements. ASM staff will contact you if anything needs to be revised before copyediting and production can begin. Otherwise, you will be notified when your proofs are ready to be viewed.

Sincerely,
Benjamin Liu
Editor
Microbiology Spectrum